# Lure Monitoring for Mediterranean Fruit Fly Traps Using Air Quality Sensors

**DOI:** 10.3390/s24196348

**Published:** 2024-09-30

**Authors:** Miguel Hernández Rosas, Guillermo Espinosa Flores-Verdad, Hayde Peregrina Barreto, Pablo Liedo, Leopoldo Altamirano Robles

**Affiliations:** 1Electronics Department, National Institute of Astrophysics, Optics and Electronics, Sta. Ma. Tonantzintla, Puebla 72840, Mexico; mhdzr@inaoep.mx (M.H.R.); gespino@inaoep.mx (G.E.F.-V.); 2Computational Sciences Department, National Institute of Astrophysics, Optics and Electronics, Sta. Ma. Tonantzintla, Puebla 72840, Mexico; hperegrina@inaoep.mx; 3El Colegio de la Frontera Sur, Tapachula 30700, Mexico; pliedo@ecosur.mx

**Keywords:** lure monitoring, TVOC index, eCO_2_ index, air quality sensors, precision agriculture, smart traps, pest monitoring, pest control, Mediterranean fruit fly, smart agriculture, smart farming, smart pest control, smart traps

## Abstract

Effective pest population monitoring is crucial in precision agriculture, which integrates various technologies and data analysis techniques for enhanced decision-making. This study introduces a novel approach for monitoring lures in traps targeting the Mediterranean fruit fly, utilizing air quality sensors to detect total volatile organic compounds (TVOC) and equivalent carbon dioxide (eCO_2_). Our results indicate that air quality sensors, specifically the SGP30 and ENS160 models, can reliably detect the presence of lures, reducing the need for frequent physical trap inspections and associated maintenance costs. The ENS160 sensor demonstrated superior performance, with stable detection capabilities at a predefined distance from the lure, suggesting its potential for integration into smart trap designs. This is the first study to apply TVOC and eCO_2_ sensors in this context, paving the way for more efficient and cost-effective pest monitoring solutions in smart agriculture environments.

## 1. Introduction

Monitoring pest populations is crucial in agriculture, due to their significant effects on global food production. Traditionally, this involves deploying special traps across fields and manually inspecting each one—a process that is time-consuming, prone to human error, and requires frequent visits by operators to visually count trapped insects. This method is not only costly and inefficient but also critical for countries that need to meet the phytosanitary requirements of fruit importing nations [1]. The effectiveness of the traps largely depends on the attractants used, which vary in their ability to lure specific species. Moreover, the use of chemical attractants and insecticides within these traps poses environmental risks if not managed carefully. Additionally, regular maintenance and monitoring of these traps are labor-intensive and expensive, particularly across extensive areas [1].

The advent of precision agriculture (PA) tools has ushered in a new era, enabling farmers to analyze the spatial-temporal variability of several critical factors that influence plant health and productivity. Data collected through sensors are stored and synthesized to guide decision-making processes and implement early warning systems to mitigate threats [2]. Additionally, there has been a shift towards *smart traps*, which enhance traditional methods by improving the accuracy of insect counting and detection capabilities. This transition is driven by the demand for more efficient, less labor-intensive techniques that allow for continuous monitoring and data collection [1].

Advancements in sensor technology and microprocessors have revolutionized the development of new devices that facilitate the detection and monitoring of insect pests over extensive areas at reduced costs and deployment times. These innovative devices, integral to early warning systems, continuously monitor the pest community and its quantitative distributions, helping to avert potential agricultural disasters [3]. Although integrating such technologies involves multiple disciplines, the benefits are substantial [3].

Modern sensors can be seamlessly integrated with the Internet of Things (IoT) or directly connected to cloud computing services, enhancing decision-making processes by enabling real-time surveillance at the field level [2]. This connectivity allows for immediate data analysis and response, crucial for managing dynamic pest populations effectively.

Moreover, the application of new technologies extends beyond traditional monitoring methods. Examples include radar technologies that track pest migration patterns [4], video equipment for observing flying insects, thermal infrared imaging to detect heat signatures, and chemiluminescent tags to track insect movements in darkness [5]. These tools not only improve the accuracy of pest detection but also contribute to comprehensive field surveys across various pest species, supporting more targeted and effective pest control strategies.

Different traps and sensors have been applied for pest monitoring. For instance, to monitor the moth *Cydia pomonella*, Guarnieri et al. [6] developed a system modifying a trap used in the field with a mobile phone to capture and report the data; the data were sent to a remote server where images were analyzed. The sound produced by *Rhynchophorus ferrugineus*, a palm tree pest, was digitally processed to generate the sound spectrum and detect its presence when the pest eats or moves [7]. Similar works based on sound recording were reported in [8,9] for detection and classification of pests. Robot cars have been used to monitor pests of Pyralidae species [2]. Liu et al. [10] mounted an image-processing system in a robot car to detect and count the number of moths in the field, reaching an accuracy of 95%. For the *Mythimna separata* species, Wang et al. [11] used a radar system to detect their presence and wingbeat, being able to detect insects with a length of 10–42 mm using FMCW radars in the W and S bands.

The Mediterranean fruit fly, one of the most critical pests in agriculture, has a high reproductive rate. When these flies lay eggs below the crop surface, the larvae feed inside the fruit, reducing its quality [12]. In 2019, losses attributed to the Mediterranean fly in Brazil amounted to USD 120 million, severely impacting exports to Japan, the USA, and Chile [13]. Generally, pest control relies on pesticides, whose efficiency depends on timely and precise location information about infestations. Unfortunately, pesticides can adversely affect natural pest enemies, beneficial insects like pollinators, contaminate water, and pose risks to human health [14]. This necessitates innovative and sustainable pest management strategies [15].

Trap monitoring for the Mediterranean fruit fly, essential in pest detection, suppression, and eradication programs worldwide, typically involves olfactory and possibly visual stimuli to attract adult species. Most attractants are food-type, emitting ammonia and simulating protein sources [16]. However, maintaining a trap network is costly, considering the required monetary, human, and material resources. Staff typically check traps weekly, traveling long distances to areas that may be difficult to access, which delays information flow [17,18]. The information on pest species and densities is primarily acquired through visual inspection, which complicates monitoring population dynamics due to the relatively low sampling rate. Smart traps address these issues efficiently by identifying and counting pests as they enter the trap, allowing for faster information flow [14].

### Sensors on Smart Traps for Mediterranean Fruit Fly

Different technologies have been applied to monitor and capture the Mediterranean fruit fly. Potamitis et al. [19] modified a McPhail trap, adapting an optoelectronic sensor to monitor the flies’ entrance by sensing wingbeat. The goal was to analyze the generated optoacoustic spectrum, reaching an accuracy of 91% in detection. An updated version of the system integrates a bimodal optoelectronic sensor and stereo recording [20], showing that it was possible to distinguish between fruit fly species (*Ceratitis capitata* and *Bactrocera oleae*) with an accuracy of 98.99%. Image capture is robust and can be used by entomologists or image-processing systems for decision-making. Therefore, camera sensors have also been used in smart traps. Doitsidis et al. [17] developed a system based on a McPhail trap modified with a camera that monitored *Bactrocera oleae* and allowed remote access to the images, reducing the time spent in collecting data. The system does not add automatic image recognition; expert entomologists analyze the images. In a similar way, Shaked et al. [16] created two systems, one to monitor *Ceratitis capitata* and another for *Bactrocera oleae*, *Dacus ciliates*, and *Rhagoletis cerasi*. Both were based on a real-time surface image sent to a remote server for image analysis, reaching 88% accuracy. Haff [21] used hyperspectral images to classify the spots of the fruit fly in mangoes. The images were analyzed using Gaussian blur radius, ball radius, and minimum particle size techniques. However, the main issue of the system is the cost and size of hyperspectral cameras, making it an impractical solution for the field. Recently, Diller et al. [5] created a surveillance system based on a McPhail trap modified with a camera and Raspberry Pi Zero for trapping and wireless transmission of images to the cloud. Based on a deep learning (DL) model, a precision of 93–95% in the species identification was reached. Similarly, Uzun [22] reported the training of deep learning algorithms to detect and count *Ceratitis capitata* in the field. The algorithm was trained with 722 images and 150 for validation. The algorithm detected and counted the species with an accuracy of 99.5%; however, no hardware was embedded in the trap for capturing images.

Other types of sensors have been integrated into smart traps, presenting innovative approaches for pest detection. Kalamatianos et al. [23] equipped a McPhail-type trap with a system that included different instruments, such as wind and temperature sensors, WiFi, and a GSM modem. The system gathered data from the field and used a public pre-trained toolkit for identifying the species *Bactrocera oleae*. With this information, an automatic classification of the species using different convolutional neural networks (CNN) reached an accuracy of 91.5%. In [24], a system based on sensors operated remotely in a McPhail trap was implemented. The capture module used a double infrared sensor placed in line to validate that the fly enters the trap through a tunnel designed to let in only Mediterranean flies. A microcontroller (MSP430F449) was responsible for processing GPS information, temperature, humidity, and wind speed; a *host control platform* (HCP) received commands through a GSM module, with information processed and stored in an SQL table using National Instruments LabView©. An improved mechanism that prevents double-counting depending on the route that flies follow inside the trap was proposed by Liao et al. [15]. The system implements nodes equipped with ZigBee modules to cover extensive and difficult-to-access areas the GSM network does not cover. In this way, the information from the nodes can be transmitted to a gateway that sends data to the HCP.

Infrared sensors have also been used to measure wing beat in McPhail traps, as proposed by Potamitis et al. [19]. The captured signal passes through an analog–digital converter (ADC), which delivers a signal in time and amplitude to which a fast Fourier transform (FFT) is applied to characterize the spectrum in frequency. With this, it is possible to identify the Mediterranean fly and different species. In a subsequent work [20], the light from the infrared LED was passed through a Fresnel lens and collimated. The wing beating of the insect casts a shadow on the opposite receiving Fresnel lens. The collimated light is partially dispersed laterally at 90° and directed to the passive Fresnel lens that records the reflected light. Finally, a dark cone-shaped plastic element fixes the LED and photodiodes to their correct focal point. A mixture of infrared and sound sensors was implemented by Sandrini et al. [13]. The infrared sensor measures wing beat, while the signal amplifier output of the sound sensor is connected to the input of a sound card. The beating signal was recorded. A time and frequency analysis determined the main components and characteristics that identify each species. Recently, Hernandez et al. [25] used a radar system to count the number of fruit flies captured in a trap. The radar system was able to detect the fruit flies using the shadow effect, which changes the radar intensity when the Mediterranean fly is inside the trap, demonstrating an efficient approach to solving pest detection.

The literature review shows that integrating different sensors strengthens the systems to monitor and detect Mediterranean fruit flies. However, research on new sensors in smart trap systems applied to fruit fly detection is still in development, and several challenges must be addressed. One of the various components of this kind of trap is the lure, which plays a vital role in the capture of insects. Commonly, the lures are based on volatile organic compounds (VOCs) that attract the insect to the trap when released into the air.

Trimedlure©, a synthetic lure consisting primarily of four trans isomers, is highly effective in attracting male Mediterranean fruit flies. This primarily consists of a mixture of four stereoisomers of tert-butyl 4(or 5)-chloro-2-methylcyclohexanecarboxylate. The most active component of this mixture is the (1R,2S)-isomer, which is responsible for the majority of the attractant’s efficacy [26]. The lure is considered superior to many other synthetic and natural attractants, owing to its unique configuration and potent behavior-inducing properties [27]. Although specific data on its longevity under various environmental conditions are not detailed, the molecule’s effectiveness is sufficiently sustained for widespread use in traps for surveying and controlling medfly populations [27]. Trimedlure© is classified as a volatile organic compound (VOC) as its structure incorporates volatile elements that allow it to evaporate and disperse into the air, a characteristic behavior of VOCs, enhancing its effectiveness as an aerial dispersant.

However, VOCs are unstable and can be degraded by environmental factors, such as temperature, humidity, and light. So, without an active lure, no capture is possible, and the trap is useless. Currently, the only information about the useful life of the lure is provided by the manufacturer based on reference conditions. Therefore, obtaining information about the lure lifetime is essential to keep the trap network in optimal conditions, according to regional and weather conditions.

This work proposed using air quality sensors to monitor and detect the Trimedlure©. Two sensors were used to show the potential of this approach: the *SGP30* and *ENS160*. The sensors measure the air concentration of VOCs (volatile organic compounds) (TVOC, Total VOC) and the air carbon dioxide concentration (eCO_2_). The integration of air quality sensors showed promising results as an aid tool that could be implemented in new smart trap systems to improve the information gathered and reduce trap visits to replace lures. To the best of our knowledge, this is the first work to monitor lures in traps.

## 2. Materials and Methods

### 2.1. Sensors and Hardware Setup

Volatile organic compounds (VOCs) are ubiquitous in both indoor and outdoor environments, with over 5000 different types identified, many of which are harmful to human health and the environment [28]. The ability to monitor VOCs accurately is crucial due to their potential adverse effects. Various sensors are available that can detect changes in gas concentrations, facilitating data collection for further analysis and decision-making. For instance, photoionization detectors (PID) utilize ultraviolet light to ionize gas molecules, allowing for the detection of VOCs by measuring the resultant charge carriers. Flame ionization detectors (FID), commonly used in industrial applications, detect hydrocarbons by burning them and measuring the ions produced. Metal oxide semiconductor (MOS) sensors detect specific compounds, like benzene, ethanol, and toluene, by changes in resistance across a thin metal oxide layer. It is important to note that some VOC sensors require temperature compensation for accurate readings, though this feature may not be integrated into all sensor models [29].

Carbon dioxide (CO_2_) is a colorless and odorless greenhouse gas. Its concentration, typically around 400 ppm in ambient air [30], can be measured using various sensing technologies. Non-dispersive infrared (NDIR) sensors determine CO_2_ levels by detecting the amount of infrared light absorbed at a specific wavelength (4.3 μm). Photoacoustic spectroscopy involves exposing a gas sample to electromagnetic energy tuned to CO_2_’s absorption wavelength, then measuring the resultant pressure waves with an acoustic detector to calculate the gas concentration. Electrochemical sensors detect CO_2_ by measuring the current change when CO_2_ reacts with a polymer surface inside the sensor. Additionally, metal oxide (MOX) technology utilizes a thin film that alters its resistance in the presence of CO_2_, providing a measure of gas concentration.

This study employed two types of metal oxide (MOX) gas sensors, the SGP30 and ENS160 from ScioSense^®^, Eindhoven, the Netherlands, designed to detect a broad spectrum of volatile organic compounds (VOCs) and equivalent carbon dioxide (eCO_2_). Both sensors operate based on the principle that the resistance of the metal oxide layer changes in response to gas exposure. Notably, the ENS160 requires a warm-up period of up to 20 min and is suitable for high-power applications, whereas the SGP30, suitable for low-power, battery-operated devices, reaches stability after just three minutes. The ENS160 also features independent hot plate control, enhancing its selectivity and sensitivity by compensating for environmental factors, such as humidity and ozone levels. In contrast, the SGP30 requires an external sensor to regulate temperature.

While various types of traps are available for capturing the Mediterranean fruit fly, for instance, the McPhail trap, our experiment exclusively utilized a modified Delta trap, also known as a Jackson trap (Figure 1a). This choice was made to integrate the sensor hardware effectively and because it is a widely used trap design in the field for Mexico [31]. As depicted in Figure 1b, the adapted trap design includes five slots at the top specifically for air quality sensor placement. These traps were constructed using a 3D printer and PLA—a biodegradable material—following the specifications from CAD models created in Fusion 360 software (v2.0.19725). The design retains the traditional dimensions of Delta traps but features a central basket. This addition is crucial as it stabilizes the lure’s position within the trap, thereby minimizing variability in the sensor readings and ensuring consistent data collection.

Figure 1b illustrates the CAD model of the trap, designed using Fusion 360 software. This model retains the standard dimensions of a conventional Delta trap but incorporates five slots at the top specifically designed for mounting the air quality sensor. In practical field applications, the lure is housed within a plastic basket equipped with a hook, forming a soft grid to secure the lure. For our experiments, a similar basket was centrally placed within the trap to prevent lure movement and minimize measurement errors. This central placement does not impact the air flow or dispersion patterns within the trap, as confirmed by Lewis et al. [32].

The trap components were produced using a 3D printer, based on the CAD model. It consists of two main parts: the body and the lid, which were assembled using screws and nuts. The material used for printing was polylactic acid (PLA), a biodegradable polymer known for its low melting point, making it ideal for such applications.

The sensors within the trap are managed by an STM32F401 microcontroller board from St Semiconductors, featuring a 32-bit ARM Cortex-M4 core running at 84 MHz, with 512 KB of flash memory and 96 KB of RAM. Communication is facilitated via an I2C interface, programmed using STM32CubeIDE software (1.13.0) in C language and utilizing the Hardware Abstraction Layer (HAL) library. This library supports the Arduino ecosystem, simplifying the development process by eliminating the need for additional programming.

#### 2.1.1. Experimental Design

The experiment is designed to assess the sensors’ capability to detect the presence of Trimedlure© (Suterra LLC, Bend, OR, USA), a widely used lure in Mediterranean fruit fly control strategies [27,31]. The primary objective is to validate the use of air quality sensors for detecting the presence of the lure. By monitoring the chemical emissions from Trimedlure©, this study aims to establish whether the sensors can accurately indicate the presence of the lure, potentially reducing the need for frequent trap visits. This approach could lead to more efficient pest management by enabling real-time monitoring of lure presence, ensuring that traps remain operational without unnecessary maintenance efforts.

Initial tests were carried out in a controlled environment—a clean room maintained at 25 °C and 20% relative humidity—to establish the baseline noise levels for the sensors. The controlled environment ensured minimal interference from external variables, providing a reliable baseline for sensor performance.

#### 2.1.2. Sensor Placement and Distance Variation

The lure was strategically placed at varying distances on the *x*- and *y*-axes, as shown in Figure 1c. For the *y*-axis, the distances were set at 1 cm, 2 cm, and 3 cm from the sensors placed in the slot 1 (Figure 1b). For the *x*-axis, the distances were set to 2 cm (slot 1), 4 cm, 6 cm and 8 cm (slot 4) from the basket in the trap lid (Figure 1b). Initial trials involved adjusting the sensor’s position along these positions and testing the sensor response to the lure. The arrows in the Figure 1c indicate the direction of distance variation for both axes, providing a clear representation of how the lure placement was varied during the experiments.

#### 2.1.3. Data Collection and Analysis

Data collection focused on monitoring changes in CO_2_ and VOC concentrations across varying distances during ten-minute intervals. Although additional time intervals were previously evaluated, the ten-minute duration was ultimately selected due to its efficiency in yielding results comparable to those from longer periods. This duration proved optimal for assessing each sensor’s response to the chemical emissions from the lure under controlled environmental conditions.

The main components of the insect pheromones, specifically VOCs and eCO_2_, were monitored using the air quality sensors. These components are critical for understanding the presence and concentration of pheromones which attract the Mediterranean fruit fly. The sensors’ ability to detect these components is crucial for evaluating the effectiveness of the lure and the potential for real-time pest monitoring.

The collected data were analyzed using statistical methods to evaluate the relationship between sensor readings and lure proximity. The effectiveness of the sensors was determined by comparing the baseline noise levels to the changes in the CO_2_ and VOC concentrations when the lure was present. Each measurement was repeated ten times, and all the samples were averaged to obtain a final value. This approach ensured the reliability and accuracy of the results by maintaining consistency across different trials. As a result, the mean error was reduced, and the confidence interval was narrowed. The final sensor readings were presented as the mean, standard deviation, and confidence intervals for each set of measurements.

Additional experiments were conducted to assess the impact of environmental factors, such as temperature, humidity, light, and airflow velocity, on sensor performance. These factors are critical as they can significantly influence the accuracy and sensitivity of the sensors in real-world conditions. During the initial tests, these environmental variables are carefully controlled to establish a baseline for sensor performance.

Subsequent tests systematically varied each factor to observe its effects on sensor readings. For example, temperature was adjusted within a range typical for field conditions to determine its impact on the sensors’ ability to detect CO_2_ and VOCs. Humidity levels were varied to assess how moisture in the air might affect sensor accuracy. The influence of different light conditions, including both natural and artificial light, was examined to understand any potential interference with sensor readings. Additionally, airflow velocity was modified to simulate wind conditions that could affect the dispersion of pheromones and, consequently, the sensor’s detection capabilities.

By systematically varying these environmental factors, the experiments provided comprehensive insights into the robustness and reliability of the sensors under different conditions. This approach ensured that the final conclusions about sensor performance are not only based on controlled laboratory conditions but also reflect potential real-world scenarios, thereby enhancing the applicability and validity of the findings.

## 3. Results

### 3.1. Sensor Performance Evaluation

The experimental results reveal distinct performance capabilities between the two tested sensors, the SGP30 and the ENS160, in detecting the presence of Trimedlure© substance using measurements of equivalent carbon dioxide (eCO_2_) and total volatile organic compounds (TVOC). The SGP30 sensor demonstrated limited effectiveness in eCO_2_ detection, showing no significant change that could reliably indicate the presence of the lure. However, it could detect variations in TVOC levels, although the data were prone to noise and influenced by external factors, such as the ambient temperature and humidity.

Conversely, the ENS160 sensor exhibited robust detection capabilities for eCO_2_ and TVOC, with less sensitivity to external disturbances. This sensor maintained consistent performance across various experimental conditions, and its readings significantly correlated with the known concentrations of lure substances. The detailed results are as follows:SGP30 Sensor Findings (Figure 2a,b):-eCO_2_ Detection: The sensor failed to show any significant change in eCO_2_ levels, remaining at baseline values around 500 ppm regardless of the lure presence.-TVOC Detection: The sensor responded to the lure presence with an increase in TVOC measurement, reaching up to 140 ppb. However, the response was unstable, fluctuating significantly with environmental changes.

ENS160 Sensor Findings (Figure 2c,d):-eCO_2_ Detection: The ENS160 showed a clear response to the lure with eCO_2_ levels increasing by an average of 200 ppm above the ambient baseline, providing a reliable indicator of lure presence.-TVOC Detection: This sensor detected TVOC concentrations consistently above the baseline, with an average increase of 300 ppb when exposed to the lure. The measurements were stable across multiple tests, with a low variance.

### 3.2. Distance-Based Sensor Performance

Further analysis focused on the effect of sensor distance from the lure on detection effectiveness. The lure was strategically placed at varying distances (1 cm, 2 cm, and 3 cm along the *y*-axis) from the sensors within the basket of the trap (Figure 1c). Initial trials involved adjusting the sensor’s position along the trap slots (*x*-axis); however, this approach proved ineffective and was discontinued in further tests. The optimal performance for the ENS160 sensor was observed at a 2 cm distance from the lure (Figure 3a,b), where both the eCO_2_ and TVOC readings were maximized and most consistent, as described below:

1 cm distance: At this proximity, both sensors showed heightened sensitivity, but the ENS160 readings exhibited a tendency towards saturation, suggesting that too close a placement may lead to overestimation of lure concentrations (Figure 3c,d).2 cm distance: This distance was found to be optimal, offering a balance between sensitivity and accuracy, with clear differentiation between baseline and lure-present states (Figure 3a,b).3 cm distance: At this range, the effectiveness of the sensors decreased slightly, with lower but still detectable increases in both eCO_2_ and TVOC levels compared to closer ranges (Figure 2c,d).

These results underscore the ENS160’s suitability for integration into smart trap designs, providing reliable, real-time monitoring of lure conditions that can significantly enhance pest management strategies.

## 4. Discussion

To the best of our knowledge, there are no existing studies that employ air quality sensors for lure detection, which precludes direct comparisons with previous work. We conducted a comparative analysis based on the readings of the ENS160 and SGP30 sensors to evaluate their respective efficiencies and capabilities. However, only the ENS160 sensor demonstrated stable and reliable performance, allowing for detailed statistical analysis. The SGP30 sensor was discarded due to instability in measurement readings.

Table 1 presents key statistics, demonstrating the superior detection capabilities of the ENS160 sensor. With a baseline noise level set at 500 ppm for CO_2_ and 40 ppb for TVOCs, the ENS160 consistently detected the lure across all tested distances. It registered average CO_2_ levels of 785.5 ppm and TVOC levels of 336.2 ppb, underscoring its high sensitivity to both gases and its ability to detect the lure’s oxidation reactions, which vary by proximity, as shown in Table 1.

In contrast, the SGP30 showed limited sensitivity, influenced by external factors like body transpiration or positional changes of the lure. The ENS160 demonstrated greater stability and less sensitivity to such disturbances, maintaining lower standard deviations of 86.1 ppm for CO_2_ and 133.6 ppb for TVOCs. This robust performance was also reflected in the consistent detection of the lure across various distances, with the optimal distance being 2 cm, where the sensor achieved the most accurate readings.

The results reveal that the ENS160 not only outperforms the SGP30 in stability and detection efficiency but also ensures reliable measurements across different settings, evidenced by a 95% confidence interval of 6.8 ppm for CO_2_ and 10.6 ppb for TVOCs, underscoring the sensor’s reliability.

The robust performance of the ENS160 sensor, particularly at a 2 cm distance from the lure, not only suggests its viability for practical applications but also highlights its potential to enhance the efficiency of pest monitoring systems. This finding could lead to the development of more sophisticated, automated trap systems that require less frequent maintenance and manual inspection. Such systems are especially valuable in large-scale agricultural settings where the costs and logistical challenges associated with traditional monitoring methods are significant.

Despite these promising results, several challenges remain. The experimental setup, although controlled, does not entirely replicate field conditions where environmental variables, such as temperature fluctuations and variable wind patterns, can affect sensor readings. Future studies should aim to test these sensors under a broader range of environmental conditions to better understand the dynamics of sensor performance outdoors.

The relationship between lure degradation and environmental factors, such as temperature and humidity, presents a promising avenue for further research. Understanding these relationships can lead to the development of predictive models that help optimize the timing of lure replacements and trap maintenance, potentially reducing costs and increasing the efficacy of pest management programs. Additionally, integrating a broader array of sensors could enhance the accuracy and reliability of these systems.

By confirming the feasibility and effectiveness of using air quality sensors in this novel application, this study contributes to the fields of smart agriculture and pest management. It opens up new possibilities for reducing labor and maintenance costs while potentially increasing the effectiveness of monitoring systems. These advancements support the broader goals of precision agriculture, which seeks to optimize resource use and management practices to improve crop production and sustainability.

In the research, we did not find related work that uses air quality sensors to detect the lure so that we cannot compare our results with other works. However, we can compare the performance of the ENS160 sensor with the SGP30 sensor.

## 5. Conclusions

In this study, two sensors, SGP30 and ENS160, were rigorously tested to assess their effectiveness for monitoring lures in traps designed to capture the Mediterranean fruit fly. The experiments demonstrated the superior performance of the ENS160 sensor over the SGP30 in detecting both CO_2_ and TVOCs, the chemical markers indicative of lure presence. This distinction was particularly pronounced at a distance of 2 cm from the lure, which emerged as the optimal position for sensor placement within the trap, providing a practical reference for trap design.

These findings underscore the ENS160’s robustness against external environmental factors, a critical advantage in field conditions where variability is prevalent. The sensor’s design and technology enable more stable and reliable monitoring of lure integrity, which is pivotal for maintaining the efficacy of traps over time. The stability of CO_2_ and TVOC readings across different sensor positions further supports using these metrics as reliable indicators of lure health, providing a foundation for predictive maintenance of the traps.

Future research should focus on integrating additional sensors to further enhance the accuracy and reliability of the monitoring system. A promising direction involves investigating the correlation between lure degradation and environmental factors, such as temperature and humidity, which could lead to the development of more adaptive and resilient pest management strategies. Moreover, forthcoming studies will involve extensive field tests to assess the long-term performance of these sensors under diverse environmental conditions and explore their potential integration into advanced smart trapping systems. These systems would not only monitor lure integrity but also provide real-time data on insect population dynamics, significantly improving the efficiency and effectiveness of pest management efforts.

By confirming the feasibility of using air quality sensors for real-time monitoring of lure conditions in pest traps, this study contributes to smart agriculture and pest management. It paves the way for more sustainable practices that reduce labor and maintenance costs while increasing the effectiveness of pest monitoring systems.

## Figures and Tables

**Figure 1 sensors-24-06348-f001:**
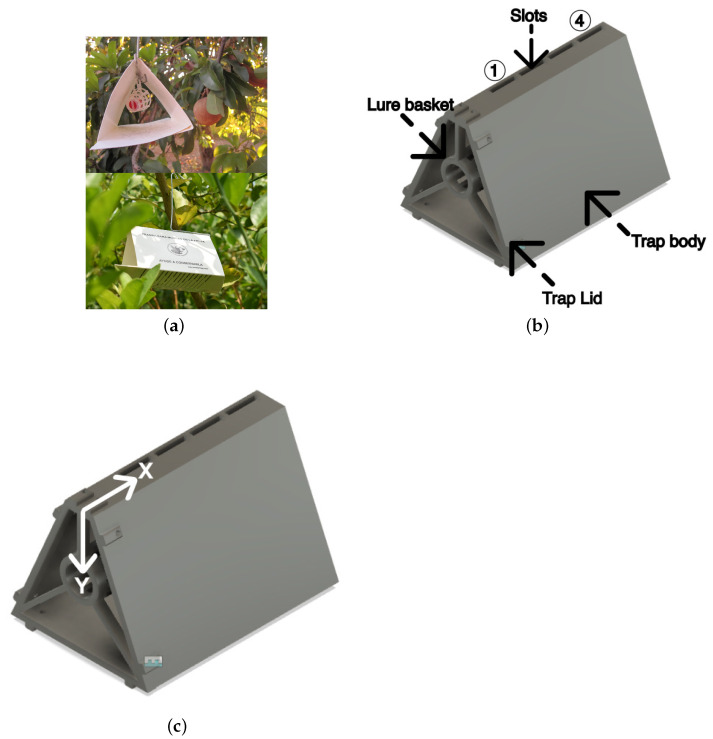
(**a**) Traditional Delta (Jackson) trap used to capture Mediterranean fruit fly, (**b**) the CAD design with *Autodesk Fusion 360*© featuring integrated air quality sensors and slots (numbered 1 to 4 along the *x* axis) for different lure placements, and (**c**) the trap with axes definition.

**Figure 2 sensors-24-06348-f002:**
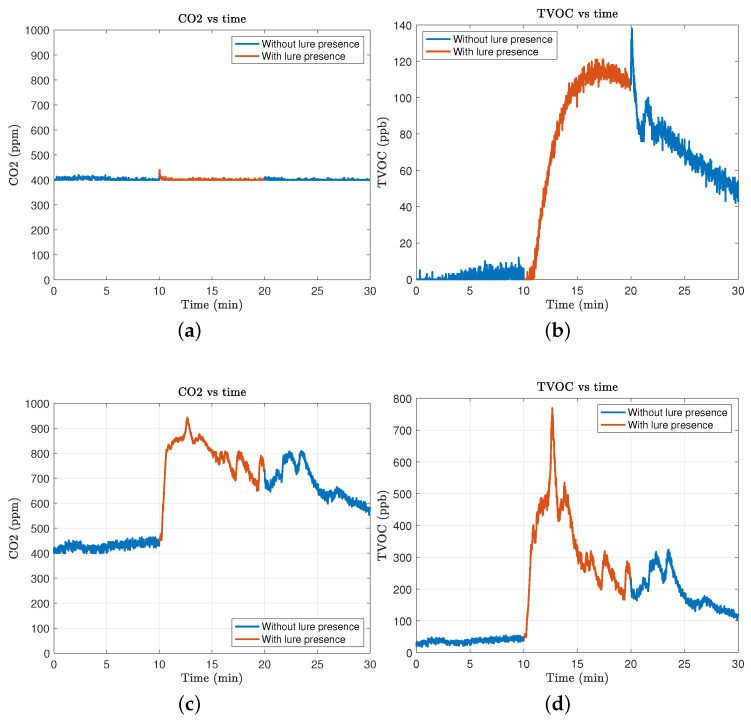
Resulting measures for the SGP30 (**a**,**b**) and the ENS160 (**c**,**d**) sensors in an experiment of 30 min: before setting the lure (min 1–10), with the lure set (min 11–20), and after removing the lure (min 21–30). The recorded data correspond to eCO_2_ (**a**,**c**) and TVOC (**b**,**d**) with a distance of 3 cm between the sensor and the lure.

**Figure 3 sensors-24-06348-f003:**
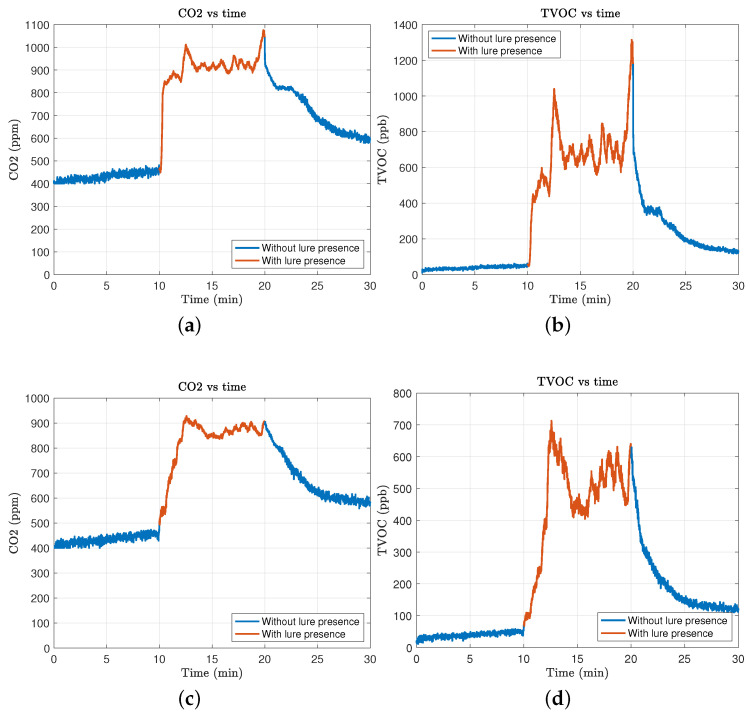
Results for ENS160 sensor when measuring eCO_2_ and TVOC at (**a**,**b**) 2 cm and (**c**,**d**) 1 cm of the distance between the sensor and the lure. Time intervals were distributed as in Figure 2.

**Table 1 sensors-24-06348-t001:** Statistics for ENS160 sensor from a ten-minute experiment for the distance of 1, 2, and 3 cm from the lure to the sensor.

Sensor Measures	1 cm	2 cm	3 cm
CO_2_			
Mean (ppm)	785.5 ± 86.1	905.7 ± 82.3	831.5 ± 97.4
Max/Min (ppm)	943/449	1075/449	928/493
Range	494	626	435
95% CI	6.8	6.5	7.7
TVOC			
Mean (ppb)	336.2 ± 133.6	665.5 ± 188.7	458.5 ± 153.4
Max/Min (ppb)	769/46	1314/46	713/68
Range	723	1268	645
95% CI	10.6	14.9	12.1

## Data Availability

The data presented in this study are available on request from the corresponding author. The data are not publicly available due to privacy or ethical restrictions.

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
