# Peer review of "Lure Monitoring for Mediterranean Fruit Fly Traps Using Air Quality Sensors"

_sensors, 2024, doi:10.3390/s24196348_

Round 1

Reviewer 1 Report

Comments and Suggestions for Authors Pest insect population monitoring is crucial in control of pests, AI and IoT is more important in the pest population monitoring. The manuscript provided a menthod for monitor insect pheromone, this is a goog method accurately to monitor the pest insect population. But the manuscript need to be improved before publication,main suggestions are as followings: 1.  the main components should be monitored with the air quality sensors, which is important for montoring the dynamic of the pest insect;  2. the model of relationship between population dynamic and changes of  the concentration of main components should be established 3.  If the air quality sensors can feel the main components of pset insects or not? 4. Why the Carbon Dioxide need to be detected?  5. How long the air quality sensors can detect the volatiles from insects? 2 cm is too short, 6. Temperature, humidity, light, airflow velocity, and other odors could affect the sensitivity of the antenna of insect? 7. Is the variation of the curve related to changes in insect populations  

Author Response

Dear Reviewer,

Please find attached a document with our point-by-point responses to your comments and suggestions. Thank you for your valuable feedback.

Best regards

Reviewer 2 Report

Comments and Suggestions for Authors

Manuscript “Lure Monitoring for Mediterranean Fruit Fly Traps Using Air Quality Sensors (authors: Miguel Hernandez Rosas, Guillermo Espinosa Flores-Verdad, Hayde Peregrina Barreto, Pablo Liedo, Leopoldo Altamirano Robles) aims to clarify the possibility of integrating two air quality sensors (SGP30 and ENS160) as an auxiliary tool for collecting information about the condition of bait in traps and, accordingly, simplifying and reducing the cost of maintenance traps as the basic tool of monitoring. The MS is read with real interest, because it is difficult not to agree with the authors that "This is the first study to apply TVOC and eCO2 sensors in this context, paving the way for more efficient and cost-effective pest monitoring solutions in smart agriculture environments." The MS well deserves publication, but needs some revision. Actually, it concerns the need to present the material more definitely and provide more clearly evidence of the advantages of the ENS160 sensor over the SGP30 one. So, at first, the authors write that “The measurements were stable across multiple tests, with a low variation” (lines 276-277), however, this statement looks completely unsubstantiated and causes confusion. Next (in the “Discussion” section), the authors begin to compare the work of both sensors and write: "we have conducted a comparative analysis of the ENS160 and SGP 30 sensors, providing valuable insights into their respective efficiencies and capabilities" (lines 298-300). However, Table 1 shows statistical data only on the ENS160 sensor, and no any sign of information about the SGP 30… And another small remark, in the caption to Fig. 2 it is necessary to indicate the distance (3 cm) at which the measurements were carried out.

Author Response

(The authors gave the same response as above.)

Round 2

Reviewer 1 Report

Comments and Suggestions for Authors

1. The manscript only provided a method about monitoring lure presence of mediterranean fruit fly, couldn't monitoring the insect population dynamics, please revise the conclusion.

2. please supplement the details about the composition of Trimedlure

3. Pay attention, Lure, pheromone and attractant. all of them are included into semiochemicals. 

4. In general, the effective duration of the lure exceeds 2 months, but the validity period of the sensor seems to be only a few minutes, please discussing in detail. 

5. How much is the cost of sensors?Is there any application value for the future? please discussing

Author Response

Dear Reviewer,

Thank you for taking the time to review our manuscript. We greatly appreciate your insightful comments and suggestions. Attached, you will find a detailed, point-by-point response to each of your comments. We have carefully considered your feedback and made the necessary revisions to enhance the clarity and quality of the manuscript.

We hope that our responses and the updated manuscript meet your expectations. Should you have any further questions or require additional information, please do not hesitate to reach out.

Thank you once again for your valuable input and for helping us improve our work.

Reviewer 2 Report

Comments and Suggestions for Authors

The updated version of the manuscript “Lure monitoring for Mediterranean fruit fly traps using air quality sensors" (authors Miguel Hernández Rosas, Guillermo Espinosa Flores-Verdad, Hayde Peregrina Barreto, Pablo Liedo, Leopoldo Altamirano Robles) contains significant corrections in the text, namely in the sections Experimental Design, Sensor placement and distance variation, Data collection and analysis, and Discussion. The corrections made significantly improve the reader's understanding of the MS's goals and objectives, as well as ensure the correctness of the conclusions. As a result, in its present form, the MS does not raise any objections and looks like it is completely suitable for publication.

Author Response

(The authors gave the same response as above.)
